# Metabolic Pathway of Natural Antioxidants, Antioxidant Enzymes and ROS Providence

**DOI:** 10.3390/antiox11040761

**Published:** 2022-04-11

**Authors:** Bernhard Huchzermeyer, Ekta Menghani, Pooja Khardia, Ayushi Shilu

**Affiliations:** 1Institute of Botany, Leibniz Universitaet Hannover, Herrenhaeuser Str. 2, 30419 Hannover, Germany; huchzermeyer@botanik.uni-hannover.de; 2Association of German Engineers (VDI), BV Hannover, AK Biotechnology, Hanomag Str. 12, 30449 Hannover, Germany; 3Department of Biotechnology, JECRC University, Jaipur 303905, India; poojakhardia.29@gmail.com (P.K.); ayushishilu11@gmail.com (A.S.)

**Keywords:** stress response, ROS, antioxidants, stress tolerance, ROS defense, futile sequences, oxidative burst, ROS signaling

## Abstract

Based on the origin, we can classify different types of stress. Environmental factors, such as high light intensity, adverse temperature, drought, or soil salinity, are summarized as abiotic stresses and discriminated from biotic stresses that are exerted by pathogens and herbivores, for instance. It was an unexpected observation that overproduction of reactive oxygen species (ROS) is a common response to all kinds of stress investigated so far. With respect to applied aspects in agriculture and crop breeding, this observation allows using ROS production as a measure to rank the stress perception of individual plants. ROS are important messengers in cell signaling, but exceeding a concentration threshold causes damage. This requires fine-tuning of ROS production and degradation rates. In general, there are two options to control cellular ROS levels, (I) ROS scavenging at the expense of antioxidant consumption and (II) enzyme-controlled degradation of ROS. As antioxidants are limited in quantity, the first strategy only allows temporarily buffering of a certain cellular ROS level. This way, it prevents spells of eventually damaging ROS concentrations. In this review, we focus on the second strategy. We discuss how enzyme-controlled degradation of ROS integrates into plant metabolism. Enzyme activities can be continuously operative. Cellular homeostasis can be achieved by regulation of respective gene expression and subsequent regulation of the enzyme activities. A better understanding of this interplay allows for identifying traits for stress tolerance breeding of crops. As a side effect, the result also may be used to identify cultivation methods modifying crop metabolism, thus resulting in special crop quality.

## 1. Introduction

Most plant species are sessile organisms. Individual plant communities can be characterized by the observed specific abundance pattern of plant species. Plant species dominating a location are optimally adapted to local growing conditions. They show a high growth rate and are able to spread and produce a high number of vigorous seeds [1]. The respective genetic potential allows plants to adapt to local growing conditions, but the growth rate is reduced if environmental factors differ from the plant’s optimal demands. Such adverse factors limiting plant performance have been termed stress [2]. Thus, stress translates into “to much” or “not enough” of the respective factor. In some cases, certain stress is comprised of more than one component. This applies to salt stress, for instance: obviously, there is an ionic component affecting the membrane potential, but there is also an osmotic component because high salt concentrations inhibit water uptake [3]. Finally, if ions enter the cytosol, they can cause a secondary effect by affecting protein structure and catalytic activity of enzymes, for instance.

With respect to terminology, we can discriminate abiotic stress exerted by environmental factors, such as light intensity, temperature, availability of water and nutrients, etc., from biotic stress originating from pathogen or herbivore attack, for instance [4]. Stress can be directly or indirectly sensed by plants. Accordingly, plants respond by initiating signaling that may directly modify enzyme activities, or the signaling cascade finally initiates regulation of gene expression. Any of these options modify physiological and biochemical parameters [5]. If the genetic potential permits adequate modifications, plants this way can adapt to an environment and continues to grow, though at a reduced rate. If stress exceeds a threshold value, the genetic potential of a plant does not allow stress adaptation, and the plant dies. Experimental approaches to analyzing the stress tolerance of plants showed that plants differ a lot with respect to the limits of stress tolerance [6].

Signaling induced by one kind of stress, such as heat, can result in modifications that are beneficial as well for a plant growing in the presence of different stress, such as water withhold. In both cases, stomata closure and improving water use efficiency are good strategies. In both cases, ABA was identified as a messenger involved in signaling [7]. However, cross-reaction of signaling pathways is not always obvious, especially if transcription factors are involved that are addressing gene families.

It is quite obvious that plants should not initiate a stress response if there are short-term fluctuating changes in growth conditions. This applies, for instance, for short-term shading and for day-night changes in illumination [8,9]. In these cases, enzyme activities are regulated by mechanisms other than stress responses, as discussed here, and carbohydrate and amino acid pools may buffer modifications in metabolite flow [10]. However, when analyzing such responses in more detail, effects were observed that were termed eustress and distress, respectively. Distress is an adverse growth condition inhibiting plant performance (as expected). Application of eustress, on the other hand, results in modifications of plant physiology and biochemistry that subsequently improve a plant’s stress tolerance. It was observed, for instance, that plants grown under slightly suboptimal conditions show an improved stress tolerance as compared to control plants grown under optimal conditions. Moreover, in several cases, this improved tolerance was also found in seeds produced by the eustress-treated plants [11]. Such observations were discussed in terms of eustress-induced chromatin modifications [12,13,14,15,16,17].

As expected, it was observed that plant performance correlated with changes in metabolite patterns and cytosolic ROS concentration, as well as the contents of antioxidants, compatible solutes and ROS scavenging enzymes. Effects based on the application of eu-stress were documented in experiments with seedlings and mature plants. It was convincingly documented that a repetitive application of moderate stress modifies metabolism, fluidity of biomembranes, and the content of ROS degrading antioxidants and enzymes [16]. The range of observed stress tolerance of treated plants correlated with the content of these ROS scavenging compounds. Economic interest was raised by the fact that (i) metabolites of pharmaceutical interest were among the compounds overproduced under eustress, and (ii) green algae and higher plants responded similarly to applied eustress [18]. Therefore it may be possible to use eustress treatments to increase the content of high-value fine chemicals in a fast-growing plant or in algae cultures in a fermenter, for instance.

## 2. ROS Production

ROS-producing reactions can be categorized into three groups: (1) ROS production as a side reaction in metabolism, (2) ROS production as a response to pathogen attack (biotic stress response), and (3) ROS production during the abiotic stress response. In any of these cases, ROS (H_2_O_2_) functions as signaling compounds in addition to potentially damaging effects. Thus, ROS control cellular homeostasis and, this way, may stimulate adaptation to adverse growth conditions (contributing to the development of stress tolerance).

### 2.1. ROS Production in Metabolism

ROS are byproducts of metabolic activity. They are important regulators of cellular homeostasis, but their synthesis and storage have to be controlled to prevent ROS from eventually reaching toxic concentrations. In all cell compartments, non-enzymatic and enzymatic antioxidants are present to protect cell contents from ROS damage. ROS results in H_2_O_2_ production and is the most important component in signaling [19].

In peroxisomes and glyoxysomes, we find oxidase activities. In leaves, the glycolate oxidase is among the most active enzymes [20]. It is involved in the photorespiration pathway (Figure 1). Further oxidase activities are linked to fatty acid oxidation (acyl-CoA oxidase) and polyamide and purine metabolism, for instance. While photorespiration-dependent oxidase activity is dominant in green tissues in the light, high rates of H_2_O_2_ production can be found in glyoxysomes of germinating oil seeds.

In chloroplasts and mitochondria, ROS are generated by electron transport activity (Figure 2). While photosynthetic electron transport and oxygen are competing for electrons released from light-activated chlorophyll in photosystem II, the Mehler reaction dominates oxygen reduction by photosystem I activity [21]. The probability of flavin-containing dehydrogenase catalyzed ROS production in mitochondria is increasing with the electron transport rate [22,23].

The NADPH-dependent oxidase is the most important source for ROS production at the plasma membrane. It has a major function in pathogen induce oxidative burst as described below. In addition, there are several peroxidases located in the cell wall. These enzymes are involved in cross-linking of cell wall polymers and, this way, are involved in cell wall hardening [24].

In the endoplasmic reticulum, protein disulfide isomerase activity produces H_2_O_2_. Two ER oxidase genes were identified in the Arabidopsis genome. Moreover, respective enzyme activity was traced in soybean seedlings [25].

### 2.2. ROS Production as a Response to Pathogen Attack

Plant defense reactions initiated by pathogen attacks are complex. Production of huge amounts of ROS is among the immediate observable effects [26]. The origin of produced ROS has been (and still is) a matter of debate. The observed effect has a lot in common with the well-known oxidative burst that can be observed after bacterial infection of animals [8]. For instance, it was observed that plant cells respond to ROS application with a hypersensitive response.

In this case, cell death can be microscopically observed in the infected area subsequent to the pathogen attack. This phenotype depends on the plant’s biochemistry rather than the activities of the pathogen. Pathogen recognition initiates ROS production in several subcellular compartments. Apparently, this response is started by ROS released out of chloroplasts, while plasma membrane NADPH oxidase only subsequently responds [27,28]. Nevertheless, NADPH oxidase activity is important for spreading hypersensitive responses from one cell to neighboring ones [29]. Hypersensitive response-related plant genes were named R genes by plant breeders because the respective activity was found to improve pathogen resistance [9]. Among the gene, functions are control of ROS production, control of autophagy, and coding for compounds of ROS signaling pathways.

The hypersensitive response can be suppressed in plant cell cultures by the addition of ROS scavenging enzymes, such as superoxide dismutase and catalase [10]. Thus, while the response can be clearly seen, results indicate that there are several ROS producers involved. Apparently, which of the potential producers are dominant in ROS production depends on plant species and the anatomical location of pathogen attack [11].

In most cases, a membrane-bound NADPH oxidase was identified to catalyze pathogen-induced ROS production [26]. In agreement with the system active in human neutrophils, O_2_^•−^ is the initial product [30]. More recent investigations using mutants of crops and Arabidopsis have contributed clarification but, at the same time, have raised further questions. Mutation of NADPH oxygenase genes resulted in an inhibition of the pathogen-induced hypersensitive response, but at the same time, they have shown that ROS are not exclusively involved in controlling cell death [31,32,33,34].

It is obvious that any pathogen defense mechanism has to be initiated by pathogen recognition. Signaling molecules such as ROS, NO, ethylene, salicylic acid, and jasmonate provoke the expression of genes and especially regulate secondary metabolism. Plant responses can be found at the infection site as well as systemically throughout the plant as a whole.

### 2.3. Abiotic Stress-Induced ROS Production

The metabolic flux, i.e., the consumption of assimilates released from chloroplasts in the light, is inhibited under stress, and the probability of light-induced ROS production increases (Figure 1) [35,36]. 

Sink regulation of photosynthetic activity is dependent on the physiological state of the plant. In other words, the plant is integrating adverse stress effects occurring at the same time on the basis of assimilating supply to sink organs [37]. Feedback inhibition of photosynthesis increases the probability of ROS production. ROS functions as messengers in retrograde gene regulation in the nucleus, but increasing ROS concentrations causes damage to cell components.

## 3. Components Controlling the ROS Level in Plants

There are two antioxidant systems in plants, pools of metabolites with antioxidative function and enzymes catalyzing scavenging of reactive species. Below we provide an overview of the most important compounds involved in plant ROS defense.

### 3.1. Major Antioxidants Involved

#### 3.1.1. Ascorbic Acid (Vitamin C)

Ascorbate is thought to be the most abundant antioxidant [38]. It interacts with other antioxidants, thus forming an antioxidant network (see: the water-to-water cycle discussed below (Figure 3) [39]. Phosphorylated sugars and nucleotide-linked sugars of the cytosolic carbohydrate pool are substrates for ascorbate synthesis, while the last steps of ascorbate synthesis take place inside the mitochondria [40]. Several pathways leading to ascorbate formation were identified, but it is generally agreed that the Smirnow–Wheeler pathway is the most important one for ascorbate biosynthesis in plants (Figure 4) [41]. In this pathway, L-galactono-1,4-lactone is the immediate precursor of ascorbate synthesis. Ascorbate synthesis is catalyzed by the enzyme L-galactono-1,4- lactone dehydrogenase. Ascorbate is not only involved in protecting from ROS damage but also has important functions in the control of the cell cycle and plant growth and differentiation, for instance [40]. Moreover, it was found as a co-factor in several enzymes [42]. Ascorbate can directly interact with ROS (H_2_O_2_, OH•, O_2_^•−^ ) and the tocopherol radical, but the most important function is the function as a substrate of enzyme-catalyzed ROS scavenging cycles (see below).

#### 3.1.2. α-Tocopherol

Tocopherols are a group of antioxidants that are present in all parts of a plant [43]. α-tocopherol (vitamin E) is the most biologically active antioxidant among the four isomers (α, β, γ, and δ). Protection of membrane lipids from oxidative damage is among the most important functions. High tocopherol concentrations are found in the chloroplasts. This is the site of tocopherol synthesis from precursors deriving from the shikimate and mevalonate pathways [44]. Surprisingly it was found that the origin of intermediates of the pathways of tocopherol and carotenoid biosynthesis changes during the maturation of chloroplasts. While young, still developing chloroplasts are autonomous, intermediates of these two pathways are imported from the cytosol into mature chloroplasts [45]. This puzzling observation was made by feeding isolated plastids with radioactively labeled intermediates. It nicely explains contradictory results on the compartmentation of enzymes involved in tocopherol synthesis because, in the experiments with Arabidopsis mutants, populations of plastids of the different developmental stages had to be used [46,47].
One hundred and twenty singlet oxygen particles can be killed by a single α-tocopherol molecule. α-Tocopherols additionally work as recyclable chain reaction terminators of PUFA (Polyunsaturated fatty acids) radicals created by lipid oxidation [48]. In this sequence, α-tocopherol first interacts with lipid peroxy radicals. Subsequently, the reaction is terminated by ascorbate to yield tocopheroxyl [49]. The most credited function of tocopherols is their contribution to different mechanisms in the protection of PUFAs from oxidation [50]. ROS produced as a byproduct of metabolism and photosynthesis are sources of lipid peroxidation in cells of plants; α-tocopherol levels were found to increase in photosynthetic tissues of plants responding to a number of abiotic stresses [51].

#### 3.1.3. Glutathione

Glutathione is a tripeptide (γ-glutamyl-cysteinyl glycine), which was found in essentially all cell compartments such as cytosol, vacuoles, chloroplasts, mitochondria, and endoplasmic reticulum [52]. It is engaged with a wide scope of processes such as cell differentiation, cell development/division, cell death, senescence, detoxification of xenobiotics, formation of metabolites, regulation of enzymatic action, synthesis of proteins, and nucleotides, lastly, articulation of stress-responsive genes [53]. The reactivity of the thiol group of glutathione makes it especially appropriate to serve a wide scope of biochemical capacities in all living beings. The nucleophilic nature of the thiol is significant in the development of mercaptide bonds with metals and for reacting with individual electrophiles [54]. For both enzymatically and non-enzymatically reduction of DHA (dehydroascorbate), GSH is used. In these reactions, two GSH molecules are oxidized to GSSG (oxidized glutathione). Glutathione reductase and NADPH are utilized for the regeneration of GSSG to recycle 2 GSH.

A focal nucleophilic cysteine buildup is responsible for the high reductive capability of GSH. Aside from being a fundamental co-substrate and reductant in protection against ROS, GSH was additionally suggested to be engaged with detecting changes in the redox poise and also sending these changes to separate target proteins [54].

#### 3.1.4. Carotenoids

Carotenoids have a place within a group of lipophilic antioxidants, which are confined in the plastids of both photosynthetic and non-photosynthetic plant tissues. They are tracked down not just in plants but also found in microorganisms. Carotenoids show their antioxidative property by securing the photosynthetic apparatus in four ways, (a) responding with LPO items to end the radical chain reaction, (b) interacting with ^1^O_2_ and creating heat as a byproduct, (c) preventing light-dependent production of ^1^O_2_ by reacting with ^3^Chl^*^ and exciting chlorophyll (Chl^*^), and (d) interacting with xanthophylls to allow transfer of excitation energy. This reaction allows the release of surplus energy as heat through the xanthophyll cycle [12].

#### 3.1.5. Flavonoids

Flavonoids are generally found in the plant kingdom preferentially in the leaves, flower organs, and pollen grains. Flavonoids can be ordered into four classes based on their structure, flavonols, flavones, isoflavones, and anthocyanins. They were considered a secondary ROS scavenging system in plants. They likewise interact with ^1^O_2_, and this way reduces the risk of peroxidation of membrane lipids [55].

Flavonoids are the only antioxidant biomolecules that possess the capacity to absorb UV radiation. Absorbed energy quanta result in a generation of ROS. The ROS generation from certain flavonoids was studied using fluorescence probes. Flavonoids generate three ROS types: the superoxide anion radical (O_2_^•−^ ), the hydroxyl radical (^•^OH), and singlet oxygen (^1^O_2_). This is based on the presence of the 2,3 double bond found in all flavonoids. In this context, the finding of Jiang et al. (2019) is noteworthy to mention: These authors published that natural flavonoids are not able to show ROS production activities. This activity rather is exerted by derivatives of quercetin such as 3′,4′,5,7-tetra-O-methylquercetin. The authors showed that this derivative is able to generate under exposure to UV and X-ray irradiation (^•^OH) and (^1^O_2_) due to the 2,3 double bond and the 3-OH group [56]. 

### 3.2. Enzymes Catalyzing ROS Removal

#### 3.2.1. Superoxide Dismutase (SOD; EC.1.15.1)

The superoxide dismutase enzyme family is arranged into three categories Cu/Zn-SOD, Fe-SOD, and Mn-SOD. They are protecting from damage by dismutating O_2_^•−^ into O_2_ and H_2_O_2_ and lessening the probability of ^•^OH formation [57]. Cu/ZnSOD is present in chloroplasts and the cytosol of the plant cell, and MnSOD is present in peroxisomes and the mitochondrial matrix. The upregulation of SODs is part of the oxidative stress response and is crucial for the survival of plants.

#### 3.2.2. Catalases (CAT; EC 1.11.1.6)

Catalases are preferentially found in peroxisomes. They are tetrameric heme-containing enzymes that convert 2 H_2_O_2_ to O_2_ + 2H_2_O [43]. Many plants have different catalase isozymes. Six were found in Arabidopsis, two in castor bean [58]. They can dismutate H_2_O_2_ or, on the other hand, can oxidize substrates such as ethanol, methanol, formic acid, and formaldehyde. Plant catalases can be grouped into three classes: class I catalases are generally noticeable in photosynthetic tissues and are associated with the expulsion of H_2_O_2_ delivered during photorespiration; class II catalases are produced in vascular tissues and may assume a part in lignification, and their accurate biological function stays obscure; class III catalases are profoundly plentiful in seeds, and young plants and their function connect with the removal of excessive H_2_O_2_ delivered during unsaturated fat degradation in the glyoxylate cycle in glyoxysomes [59].

During stress, catalases can directly dismutate H_2_O_2_; thus, they are crucial for ROS detoxification [60]. This is correlated to an expansion of areas of high abundance of peroxisomes during stress. It is thought to help scavenge H_2_O_2_ that diffuses from the cytosol [61]. Class II catalases were principally considered to correspond to disease development. It was viewed as an objective for SA (salicylic acid) signaling. For instance, the expression of the Cat2Nt gene in transgenic potato plants improved the protection of these plants from *Phytophthora infestans* [62]. Nonstop waterlogging in Carrizo citrange and Citrumelo CPB 4475 showed that CAT movement expanded 1.7 and 3.0 times compared to the control plants [63]. Harinasut et al. [64] showed that CAT action did not react to expanding salt concentration in salt-tolerant mulberry cultivar, pie. The decline in CAT action in leaves of *Bruguiera parviflora* under NaCl stress was additionally noticed by Parida, Das, and Mohanty [65]. The diminishing CAT activity in certain plants mirrors the significance of peroxidase, just as the SOD/ascorbate–glutathione cycle as oxygen reactive scavenging framework [64].

#### 3.2.3. Ascorbate Peroxidase (APX; E.C. 1.1.11.1)

APX is an essential part of the Ascorbate–Glutathione (ASC-GSH) cycle. While CAT preferentially scavenges H_2_O_2_ in the peroxisomes, APX fills a similar role in the chloroplast and cytosol. Ascorbic acid is used as a reducing agent to reduce H_2_O_2_ to H_2_O and also DHA. The APX family comprises at minimum five distinct isoforms, including thylakoid and microsomal membrane-bound structures, just as dissolvable stromal, cytosolic, and also apoplastic enzymes [66,67,68]. APX is a more efficient scavenger of H_2_O_2_ in times of stress because it is widely distributed and has a better affinity for H_2_O_2_ than CAT. The isoform that is more responsive to light-mediated oxidative stress is APX1. This is due to the suppression of tylEX. An enhanced stress tolerance could be observed when the expression of tylAPX was stimulated [68]. Wang, Zhang, and Allen (1999) tracked down that overexpression of Arabidopsis peroxisomal ascorbate peroxidase gene in tobacco provides tolerance against the herbicide aminotriazole, which causes oxidative stress in peroxisomes and glyoxysomes. However, the same transgenic tobacco plants did not show a decline in damage brought about by paraquat, which causes the development of ROS in the chloroplast [68]. They theorized that resistance given by the expression of peroxisomal APX might be explicit to the oxidative stress caused in peroxisomes rather than in chloroplasts. The expression of Cu/ZnSOD and APX genes in leaves of transgenic sweet potato plants were essentially instigated by Methyl Viologen treatment.

#### 3.2.4. Monodehydroascorbate Reductase (MDHAR; E.C. 1.6.5.4)

MDHAR is liable for recovering ascorbic acid (AA) from the fleeting MDHA, involving NADPH as a reducing agent, ultimately renewing the cell AA pool. Since it recovers AA, it is co-localized with the APX in the mitochondria and peroxisomes, where APX rummages H_2_O_2_ and oxidizes AA in the process [69]. MDHAR has a few isozymes that are bound in chloroplast, glyoxysomes, mitochondria, cytosol, and peroxisomes.

#### 3.2.5. Glutathione Peroxidases (GPX, EC 1.11.1.9)

Glutathione peroxidases are a group of numerous isozymes which catalyze the reduction in H_2_O_2_ [70,71,72]. It assumes an essential part in the biosynthesis of lignin, just as guards against biotic stress by debasing indole acetic acid (IAA) and using H_2_O_2_ all the while. GPX favors fragrant aromatic compounds such as guaiacol and pyrogallol [73] as electron donors. GPxs in plants are characterized into three kinds: selenium-subordinate (GPx, EC 1.11.1.19), the non-selenium-subordinate phospholipids hydroperoxide GPx (PHGPX), and glutathione transferases (GST, EC 2.5.1.18) showing GPx movement (GST-GPx). Due to its presence in cytosol and vacuole, it is considered a vital enzyme in the evacuation of H_2_O_2_.

PHGPx was displayed to react to salt stress [71], and this increment in action was seen in catalase deficient tobacco plants [70]. In citrus, PHGPx protein and the gene encoding were isolated and characterized. 

## 4. Regulating ROS Concentrations in Plant Cells

It was a puzzling observation that overproduction of reactive oxygen species (ROS) is a typical stress response independent of the kind of stress applied to a plant. This attracted the interest of breeders [13]. Crop stress tolerance might be achieved by either reducing the ROS production rate or increasing a plant’s capacity to remove ROS. It was generally understood that under stress ROS production rate exceeds the one of ROS removal [14]. Moreover, experimental results indicated that the rate of ROS production was stimulated under stress, while the rate of ROS removal initially remained unchanged. The question was about how ROS are generated and what are the main pathways of ROS removal. These questions were addressed in the publications reviewed below. We focused on (1) the main pathways of ROS production in plants, (2) typical antioxidants protecting plant cells from spells of sudden ROS production, and (3) enzymes catalyzing ROS degradation. These enzymes are the most important components because they allow the development of equilibrium between ROS production and consumption rates. Antioxidant molecules, on the other hand, form a pool that is consumed within a short period of time while the replenishing rate is slow.

Numerous research projects aimed at improving plant stress tolerance while keeping in mind the threats of global climate change to agricultural production of food, feed, and biomass [74,75]. Therefore, we need to better understand the regulation of catalytic activities of ROS degrading enzymes. This requires analysis of two aspects, (i) stress perception and (ii) signaling resulting in a regulation of individual catalytic activities.

### 4.1. Stress Perception and Signaling

Several forms of abiotic stress are indirectly linked to water deficit stress. This applies to ionic stresses (salt/sodic stress, high nutrient stress, etc.) and cold stress, for instance [76,77]. As a result, responses of cell turgor are observed, and the cellular concentration of ABA increases [78]. ABA-dependent and ABA-independent responses were described [79]. Due to ABA transport, respective effects can be observed in the whole plant. Stomata closure is among the most obvious ABA responses. By using molecular methods, the expression of ABA-responsive genes was observed [65,66,67,68,69,70,71,72,73,74,75,76,77,78,79,80]. Gene products can be categorized into compatible solutes and compounds conferring protection from osmotic and ionic damage. As shown in Figure 5, metabolites of primary carbohydrate and amino acid pathways function as substrates for the respective catalysis of these compounds. Among the second type of products are signaling compounds and transcription factors regulating further genes [81,82,83].

Unfortunately, the problem of the identification of stress sensors has not been completely solved. Though, it is speculated that stress sensing and signaling at the cellular level resembles the concepts known from yeast and animal cells [81]. We focused here on drought and salinity sense because these are most important with respect to problems currently caused by the phenomenon of global climate change (Figure 6 and Figure 7). Moreover, both types of stress cause most of the yield losses in agriculture [84]. Several membrane receptors are supposed to be osmo sensors responding to cell turgor. One such receptor is Cre1 (cytokinin response 1), a histidin-kinase linked receptor transducing signals via the MAP kinase pathway. As the name of this receptor already tells, it is also cytokinin-responsive [85]. Experiments with germinating seeds indicated that water uptake during germination is controlled by the osmo sensor AWPM19. This observation was proven by experiments using Arabidopsis and Arabidopsis defect mutants. Similar to the yeast osmo sensor Sho1, AWPM19 osmo signaling involves a MAP kinase signal transduction [86]. Moreover, based on in silico studies, a family of 15 homolog Ca channels expressed by the OsCa1 family of genes was identified [87,88]. These channels respond to osmotic stress but are not ABA-responsive [89].

Stress perception results in signaling on cellular, tissue, and whole plant level. Perception of ionic stress, as well as binding of signaling molecules (released from other signaling events), often starts with binding to a membrane-bound receptor. ABA that was produced upon stress sensing binds, for instance, to the receptor PYR/RCAR, which is quite common to plant tissues [90]. This receptor releases signaling via a serine/threonine kinase (SnRK2). In addition to serine/threonine-like kinases, G-protein-coupled receptors, and ion channel-like receptors, were found in most plant species analyzed so far [76]. Several signaling pathways are interlinked by receptors responsive to signaling compounds released by another respective pathway.

This applies to Ca signaling that may activate ion channel-like receptors, or cytokinins binding to the putative osmo sensor Cre1, for instance [38]. This receptor releases signaling via a serine/threonine kinase (SnRK2) [91]. In addition to serine/threonine-like kinases, G-protein-coupled receptors, and ion channel-like receptors, were found in most plant species analyzed so far [81].

### 4.2. ROS in Signaling Events

Reactive oxygen species are not only toxic side products of aerobic metabolism but also are important signaling compounds tuning metabolism as well as plant development. There is a tightly knit network of ROS signaling pathways, Ca signaling, and ethylene signaling. During optimal growth conditions, ROS synthesis rate and ROS degradation rate are balanced, allowing a constant cellular ROS level. ROS scavenging occurs by interaction with antioxidant molecules as well as enzyme-controlled degradation at the expense of the cellular ascorbate/dehydroascorbate and glutathione (GSH/GSSG) redox pools [92].

As expected, a detailed analysis of stress responses has shown that an initial more general stress response becomes more specific over time. This may be explained by the fact that initially, the stress response relays on the enzyme equipment that has proven optimal in the absence of stress [93]. Stress adaptation requires down-regulation, and degradation of respective enzymes, of some metabolic pathways, while an abundance of other enzymes has to be increased to allow up-regulation of other functions. In experiments with Arabidopsis, a large number of stress-responsive genes were identified. Some of them were responding to several different types of stress, whereas others appeared to be stress-specific. Among these genes, several are encoded for transcription factors. When the transcriptome of Arabidopsis was analyzed, 104 osmotic stress-responsive transcription factors were identified [94]. As transcription factors can control the expression of more than one gene, this explains why stress response is a very complex trait [95].

It is obvious that the sketched mechanism of stress adaptation is comprised of two parts: de novo synthesis of enzymes and metabolites and degradation of surplus components. This degradation of cell components occurs by autophagy. In addition to the removal of unwanted material, a vital function of autophagy is the recycling of organic material as a substrate for essential cell functions. Autophagy activity is strictly controlled by several factors, including messengers and final products of signaling pathways, such as ABA, ethylene, and ROS [96]. Plant species do not only differ in metabolic activity and the pattern of expressed genes. They also differ in the activity of autophagy and the scale of its regulation under stress. This situation is complicated by the fact that the extent of an expressed stress response depends on the physiological status of a plant prior to the stress event. Moreover, it was documented that this response may be further modulated by priming techniques [18].

In this context, the importance of autophagy often is underestimated. During the stress adaptation phase, plant metabolism can be significantly supported by substrates that are recycled by autophagy. Thus, autophagy compensates for losses that are caused by the inhibitory effects of the respective stress. In this adaptation phase, stimulation of autophagy by ROS is significantly contributing to ethylene-mediated stimulation. On the other hand, an overshoot has to be down-regulated to prevent a hypersensitive response that finally might induce apoptotic cell degradation. This is achieved (i) by autophagy-mediated degradation of enzymes involved in ROS production (see Table 1 and Figure 5) and (ii) by ROS scavenging enzymes.

### 4.3. Regulating the Activity of ROS Scavenging Enzymes

Typically it is observed that the activity of ROS scavenging enzymes increases as a response to environmental stress. When comparing genotypes differing in the degree of stress tolerance, the more tolerant genotype shows a more pronounced increase in ROS scavenging activity as compared to the more sensitive one [97]. It was also generally observed that the expression and activities of antioxidant enzymes not only differ between roots and shoots but also vary during the phase of stress adaptation [98].

#### 4.3.1. Catalase (CAT)

Due to its high turnover rate, catalase is limiting the cellular H_2_O_2_ concentration under optimal growth conditions. On the other hand, mobility of H_2_O_2_, along with the substrate affinity of catalase, allows an H_2_O_2_ level sufficient for its messenger function [99,100]. Nevertheless, catalase activity, especially when expression of the CAT1 gene is regulated, may modulate or even terminate H_2_O_2_ signaling [101]. Several abiotic stresses initiate a sudden burst in ROS production resulting in the H_2_O_2_ concentration eventually passing a threshold level and causing a risk of ROS mediated damage. Such events require up-regulation of gene expression to increase the cellular catalase level. Such stimulation of catalase production can be initiated by ABA. In experiments with Arabidopsis, it was clearly shown that ABA-responsive expression of CAT1 is brought about by MAP kinase (mitogen-activated protein kinase) signaling involving AtMPK1 and AtMPK6 [101]. It was observed in experiments in field studies (using *Amaranthus* and Cassava (*Manihot esculenta*), for instance), as well as Arabidopsis, that H_2_O_2_ producing SOD activity as well as H_2_O_2_ consuming CAT activity is up-regulated under stress [102,103]. Interestingly it could be shown by analysis of cassava accessions differing in drought tolerance that such differences are due to the pool size of antioxidant compounds rather than the expression and activities of ROS scavenging enzymes [103]. In other plant species, such as Cerasus humilis, differences in drought tolerance of accession apparently are linked to the activity of the ascorbate/glutathione cycle rather than the CAT activity [104].

#### 4.3.2. Superoxide Dismutase (SOD)

Hydroxyl radicals are among the most reactive ROS species able to cause denaturation of proteins, lipid peroxidation, and mutation of DNA. Sources of superoxide radicals are reactions allowing a misdirection of electrons to oxygen, such as electron transport reactions in mitochondria and chloroplasts [105]. Due to the extremely short half life of hydroxyl radicals, they are no signaling molecules, but H_2_O_2_ produced by degradation reactions has a regulatory function [106,107]. SOD is an important scavenger of hydroxyl radicals because it reacts at an almost diffusion-limited rate to produce hydrogen peroxide [105]. In leaves, chloroplastic SOD is the most abundant SOD, while in non-green plant tissues, mitochondrial SOD is most abundant. In general, it was observed that SOD encoding genes are not co-regulated but are responding to cell compartment-specific ROS production [105]. H_2_O_2_ production can be stimulated by ABA. It was convincingly shown by several teams that a MAP kinase pathway is linking ABA signaling to stimulation of gene expression and, thus, SOD synthesis [108]. In experiments with Arabidopsis, it was shown that ABA-mediated expression of the CAT1 gene involves signaling via a MAP kinase pathway. The functioning of this pathway and ABA-induced H_2_O_2_ production depends on the expression of AtMPK1 and AtMPK6 [101]. In experiments analyzing SOD production as a response to cold stress and salt stress, respectively, gene expression was stimulated by a MAP kinase signal transfer following the route MEKK1, MKK2, MPK4/MPK6 [109]. Fine-tuning of ABA-mediated SOD production and subsequent H_2_O_2_ production may be brought about by feedback inhibition. However, it also can be speculated that there is a risk of overshoot because it was observed that H_2_O_2_ could activate the MAP kinase pathway [110,111].

It was an important and quite surprising finding that there are similarities in stress responses of plant and animal cells. Heavy metals are well known for their adverse effects on cell homeostasis in cells of plants and animals. This allows discussing general concepts underlying stress tolerance in cells differing in physiology and structure as much as plant and animal cells. We mention here signaling events that were observed subsequent to exposure to cadmium, for example. Most obviously, we can find common signaling components with similar functions in both cell types. The most important ones are MAP kinases and ERK kinases (extracellular signal-regulated kinases). These kinases are involved in the regulation of important functions, such as the control of cell division. Disturbance of such pathways may lead to cell death or uncontrolled cell division, as observed in tumors, for instance. Signal transduction can be initiated by Cd sensing and subsequent release of signaling molecules such as Ca^2+^, ROS, and NO. On the other hand, significant differences were found with respect to the compounds produced subsequent to signaling-activated gene expressions, such as specific animal and plant hormones. Thus, the general concept of Cd sensing and initial response may be evolutionarily conserved [112]. This assumption is supported by more detailed observations. Both in animal and plant cells, ROS production is catalyzed by a Cd-activated NADPH oxidase [113].

Further, it became evident from the experiments of Demidchik et al., 2003, 2007, [114,115] that cell signaling is mediated by free radical and redox mechanisms. This signaling targets ion channels and receptor complex systems that start a series of responses regulating cellular metabolism. During the last decades, it became evident that in plants and animals, the production and increased content of ROS also triggers signaling pathways. These signaling events initiate responses as different as hormone synthesis geotropism and stress responses. In a similar way, Ca^2+^ signaling is involved in several signaling sequences. ROS stimulated the opening of Ca^2+^ channels can result in an amplification of stress signals. Moreover, the involvement of both signaling molecules in several signaling pathways is understood to be the reason for observed cross-reactions and reciprocal modulation of responses when several stresses are occurring simultaneously.

According to Harman’s hypothesis, mitochondria and other metabolic pathways regulating organelles are the sources of ROS production. These ROS, if produced in more than the quantified limit, lead to aging and age-related disease, which was experimentally established. This implies that there is a direct link between metabolic activity and ROS production. Due to its relative stability and mobility, H_2_O_2_ is functioning as a messenger molecule. Any disturbance of balanced metabolic activity results in the overproduction of ROS and subsequent stress response. The observed extent of signals varies among organelles. The same holds true for the signaling molecules and their targets, such as Ca_2_ + channels starting a signaling cascade, stress-activated protein kinase (SAPK) and MAPKs, and c-Jun NH_2_-terminal kinase (JNK), p38, p53 gene-related stress at the end of such a sequence. As a final symptom of stress perception, aging may be observed in plant or animal cells [116,117].

## 5. Conclusions

In oxidation, there is a deficiency of electrons or an upgrade in the oxidation condition of a particle such as ions, etc. Enzymes (oxidases/reductase) play a catalytic role in metabolic pathways resulting in a product from precursors. These metabolic pathways are the soul of any plant system. Plants in abiotic or biotic stress conditions can face oxidation or generation of free radicals. These free radicals are dangerous and can cause degenerative diseases due to a direct attack on membranes, cause mutations by DNA damage, etc. In order to cure this kind of oxidative stress, plants possess natural antioxidants, such as ascorbic acid (Vitamin C), α-tocopherol (Vitamin E), glutathione, carotenoids, and flavonoids. These natural antioxidants have the potentials to entrap free radicals. Their synthesis is based on substrates of primary and secondary plant metabolism. Thus, they are naturally present in all plants, but the concentration level may differ according to compartmentation and activity of respective metabolic pathways. Moreover, stress can alter these activities, and partition among metabolite flows affects plant growth and production.

It is evident that any type of stress (biotic and abiotic) is the primary cause of reduced plant growth and development. Therefore, recent research projects are focusing on elucidating the physiological, biochemical, and molecular basis of the stress response. The final goal is to avoid stress-induced damage and thus enhance agricultural production. Initial research described the toxic nature of ROS, but more recently, the focus has broadened. ROS now are seen as well as an essential regulatory part of metabolism. It became obvious that ROS play a critical role as messengers controlling plant growth, development, and survival in a stressful arena. In this review, we summarized that ROS could be generated (i) as side products of metabolism, (ii) as a response to pathogen attack, and (iii) as a response to abiotic stress. Moreover, we exemplified the functions of the most important antioxidants and enzymes involved in ROS production and scavenging.

We commented on the regulation of the ROS concentration in plant cells. It is important to understand the individual concept of a plant’s underlying regulation of enzyme activities and to result in the observed stress tolerance. In this context, emphasis is laid on stress perception and subsequent signaling events regulating metabolic activities as well as ROS production and scavenging. The activity of ROS scavenging enzymes increases as a response to environmental stress. When comparing genotypes differing in the degree of stress tolerance, the more tolerant genotype shows a more pronounced increase in ROS scavenging activity as compared to the more sensitive one. The abundance of antioxidant enzymes varies in the cytosol, mitochondria, plastids, and peroxisomes. In plastids and mitochondria, ROS generation can be interpreted as a failure of the electron transport system. In the other cell compartments, ROS are side products of the metabolism but can be produced by stressors as well. Therefore, more studies are required for a better understanding of stress mechanisms and their effects on metabolic pathways. What are the exact mechanisms causing the over-production of ROS resulting in the observed degenerative disease? What are the parameters limiting stress tolerance, and what is their scaling for an individual plant species? Maybe these questions can be approached by the use of AI, advanced imaging techniques, and mapping of extranuclear DNA.

As a final comment, we would like to draw attention to the described similarities in stress perception and subsequent signaling found in animal and plant cells. Similarities can be identified when also looking for reasons for ROS generation. This applies to many habits in day-to-day life, leading to exposure to stressors such as smoke, ultraviolet light, heavy metals, etc. Resulting cellular malfunctioning results in ROS generation (i.e., ROOH, -RO, -OH, O_2_, H_2_O_2_, ROO^−^, NO_2_, and ONOO, etc. In animals, we can sometimes prove that malfunctions lead to inflammation and diseases such as cancer, diabetes, neurogenerative, and/or cardiovascular malfunctioning. Therefore it may be speculated that biochemical mechanisms reducing the risk of ROS overproduction in plants may also be beneficial in animal tissues. This reasoning is the basis for ongoing research testing antioxidative plant extracts for disease treatment in animals and men.

## Figures and Tables

**Figure 1 antioxidants-11-00761-f001:**
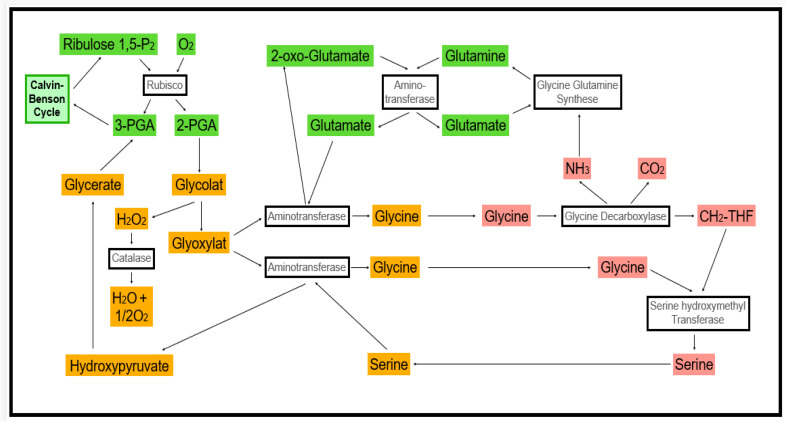
Metabolic interaction of cell compartments during photorespiration.

**Figure 2 antioxidants-11-00761-f002:**
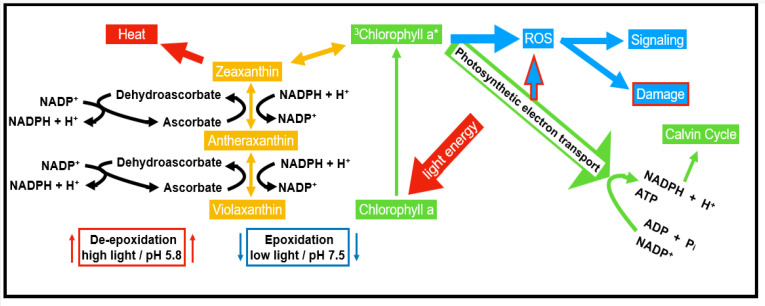
Absorption and display of energy in reaction centers and light-harvesting complexes. Chlorophyll a* in its triplet state is sufficiently stable to allow electron transfer reactions to redox components of the electron transport chain.

**Figure 3 antioxidants-11-00761-f003:**
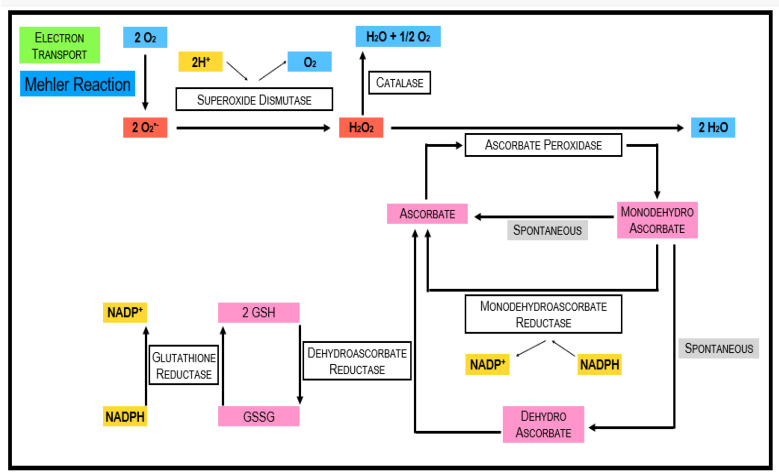
Scavenging of ROS. The figure demonstrates two alternative pathways to scavenge ROS produced by the Mehler Reaction: (i) the SOD-CAT pathway (top left) and (ii) the interplay of antioxidant enzymes in the ascorbate–glutathione cycle.

**Figure 4 antioxidants-11-00761-f004:**
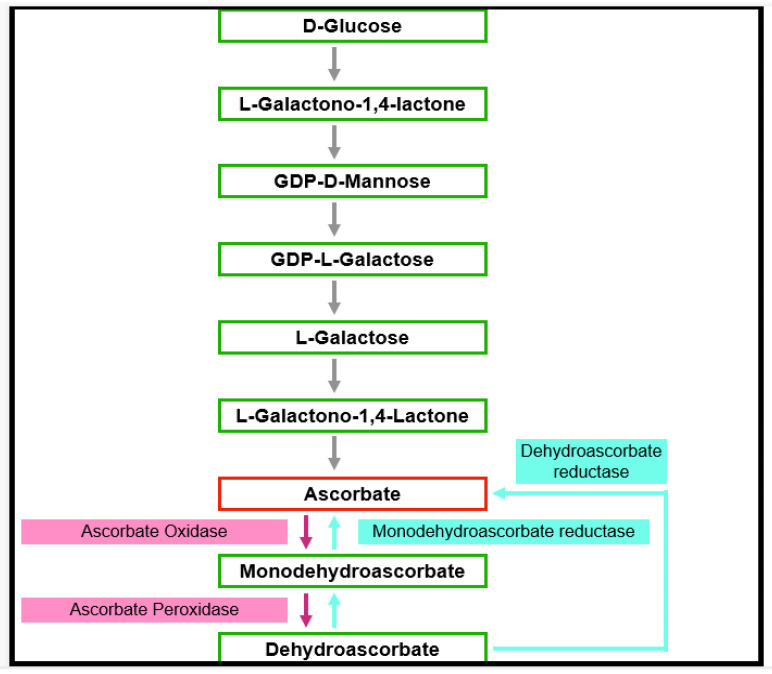
Outline of the pathway of ascorbate biosynthesis in plants. Ascorbate is a major antioxidant found in all plant tissues. The pathway of biosynthesis may vary due to the availability of precursors.

**Figure 5 antioxidants-11-00761-f005:**
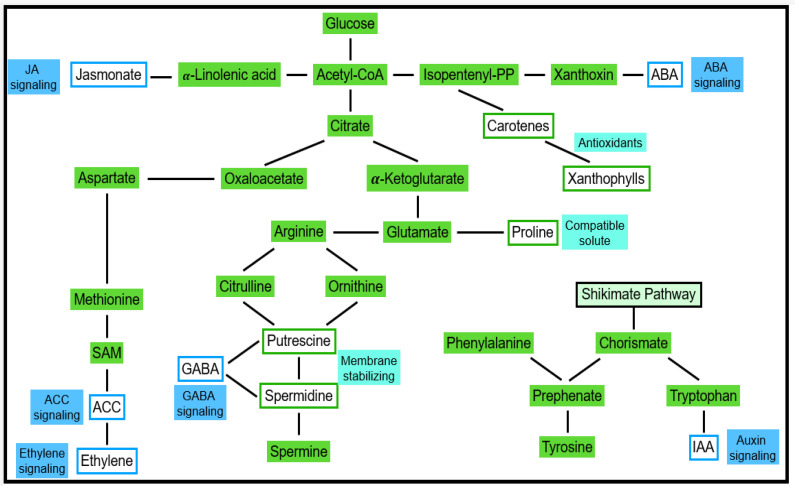
The synthesis of compatible compounds and signaling molecules integrating into plant metabolism. Depending on the respective genetic potential, plants differ in the expression of metabolic pathways. Moreover, these preferences may vary during a plant’s life cycle. This has consequences for the preferences to produce individual compounds such as compatible solutes, hormones, and other signaling molecules. Levels of these molecules depend on both availabilities of substrates (precursors) and the activities of enzymes involved in biosynthesis.

**Figure 6 antioxidants-11-00761-f006:**
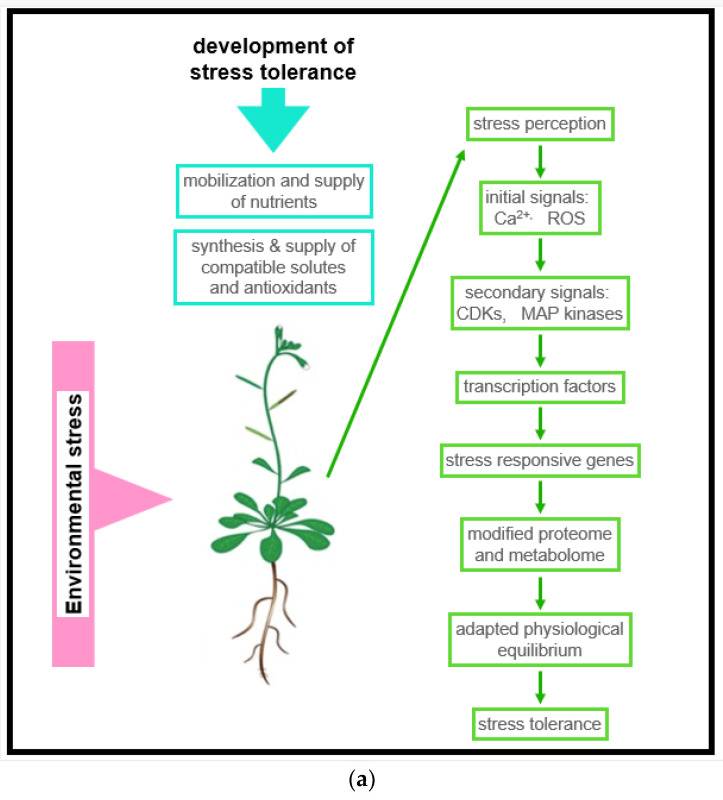
(**a**) A general concept of the reaction sequence leading to stress tolerance When analyzing plant responses to moderate drought and salt tolerance, a general scheme of responses was found. For successful adaptation to a stress event, it is important for a plant to find access to nutrients during the adaptation phase. Moreover, it is essential to control stress-induced ROS production, prevent an overshoot, and down-regulate ROS production at the end of the adaptation phase; (**b**) Example of stress signaling. Signaling intensity can be reduced (i) by ameliorating effects of compatible solutes (=reduction in the stress level), and (ii) by direct interaction of plant metabolites with the signaling molecule species or by inhibition of the synthesis of the signaling molecule (=reducing the amount of free signaling molecules).

**Figure 7 antioxidants-11-00761-f007:**
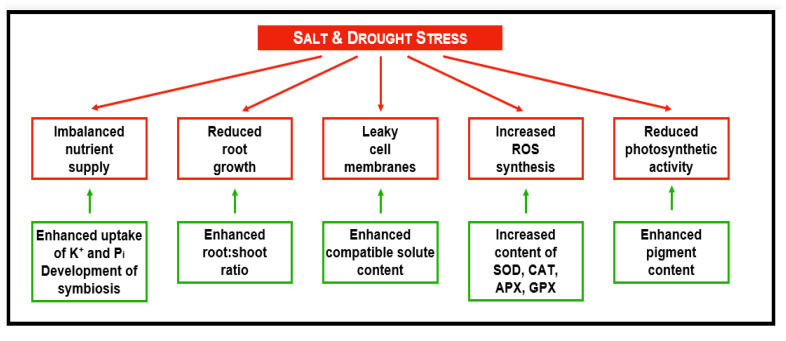
An overview of responses to salt and drought stress. Responses observed under stress are boxed in red. Green boxes are listing strategies successfully used by stress-tolerant accessions of investigated plant species to avoid adverse effects of stress. Not all of the listed strategies are used by an individual plant. The preferred response of a plant species depends on the respective genetic potential. Moreover, responses may be scaled depending on the developmental and nutritional state of a plant when exposed to stress.

**Table 1 antioxidants-11-00761-t001:** Subcellular localization of major antioxidant enzymes in plants.

Enzyme	Abbreviation	Localization
Ascorbate Peroxidase	APX	cytosol, mitochondria, plastids, peroxisomes
Catalase	CAT	peroxisomes
Dehydroascorbate reductase	DHAR	cytosol
Glutathione reductase	GR	cytosol, mitochondria, plastids
Glutathione peroxidase	GPX	cytosol, mitochondria, plastids
Monodehydroascorbate reductase	MDAR	cytosol, mitochondria, plastids
Superoxide dismutase	SOD	
	Cu/Zn-SOD	cytosol, mitochondria, plastids
	Mn-SOD	mitochondria
	Fe-SOD	plastids

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
