# Peer review of "Metabolic Pathway of Natural Antioxidants, Antioxidant Enzymes and ROS Providence"

_antioxidants, 2022, doi:10.3390/antiox11040761_

Round 1

Reviewer 1 Report

The purpose of this manuscript is to emphasize the role of oxidative stress in plant cells and the subsequent signaling events that regulate metabolic activities, as well as the production of ROS and their scavenging. The scientific collect is very interesting, however, some problems, as indicated below, should be addressed before the document can be considered for publication in this journal. This version of the manuscript is not enough complete. Here, I present all my objections in details.

Minor revision:

English language and style are not exhaustive, a greater spell check is required to ensure that an international audience can clearly understand your text. In general, I suggest reviewing the style of the manuscript according to the guidelines of the journal.

Line 27, “the enzyme” should be modified in “enzyme”

Line 40, “optimal” should be modified in “optimally”

Line 47, “a certain” should be modified in “certain”

Line 49, “potential. But” should be modified in “potential, but there is also”

Line 60, “the plant” should be modified in “plants”

Line 62, “a stress” should be modified in “stress”

Line 66, “result modification” should be modified in “result in modification”

Line 79, “disstress” should be modified in “distress”

Line 109, “(=contribute” should be modified in “(contributing to”

Line 112, “. But” should be modified in “, but their synthesis”

Line 114, the meaning of the sentence is not clear

Line 137, the acronym ER was not written in full

Line 212, there is a repetition of the word "function"

Line 137, the acronym PUFA was not written in full

Line 251, “GSSO” should be modified in “GSSG”

Line 256, “non-photosynthetic and non-photosynthetic” should be modified in “photosynthetic and non-photosynthetic”

Line 261, “moment” should be modified in “property”

Line 269-273, there are no spaces between words

Line 280, the localization of Fe-Sod should be specified

Line 298, unify the wording: class 2 or class II

Line 311-330, there are no spaces between words

Line 332, the acronym AA was not written in full

Line 339-349, there are no spaces between words

Line 428, “GSSG/GSH” should be modified in “GSH/GSSG”

Line 485, “Innterestingly” should be modified in “Interestingly”

Line 486, what is the meaning of “cassava”?

Major revision:

Regarded as the exceptional ability of flavonoids to counteract the damage related to the increase of ROS, I suggest to improve paragraph 3.1.5.

I find very interesting the observation regarding the similarity of the stress perception and its consequent signaling between plant and animal cells. Could the authors provide more information about the pathways involved and their consequences? I might suggest adding the following reference: (doi: 10.1007/s12079-012-0173-3), in order to discuss also these aspects.

Moreover, it might be interesting to point out similarities and differences in the action of ROS on ion channels of plants and animals. I might suggest adding the following reference: (doi: 10.1242/jcs.00201; doi: 10.1111/j.1365-313X.2006.02971.x; doi: 10.1111/apha.13796), in order to discuss also these aspects.

Among the consequences of ROS increase in animal cells, authors identify some pathologies. I suggest the addition of the aging mechanism related to oxidative stress, the common denominator of several diseases. In this regard, I propose to include a brief comment on the plant (doi:10.1007/s004250100646) and animal cell aging (doi: 10.1002/jcp.30632 )

Author Response

Dear Sir,

Kindly find enclosed herewith the reviewers suggestion and incorporation as per our best efforts. The authors are grateful to the reviewer for the following important suggestions.

Response to Reviewer 1 Comments

Point 1:

Line 27, “the enzyme” should be modified in “enzyme”

Line 40, “optimal” should be modified in “optimally”

Line 47, “a certain” should be modified in “certain”

Line 49, “potential. But” should be modified in “potential, but there is 

Line 60, “the plant” should be modified in “plants”

Line 62, “a stress” should be modified in “stress”

Line 66, “result modification” should be modified in “result in modification”

Line 79, “disstress” should be modified in “distress”

Line 109, “(=contribute” should be modified in “(contributing to”

Line 112, “. But” should be modified in “, but their synthesis”

Response 1: All the suggestions are incorporated and highlighted in green in the revised manuscript for your kind look

Point 2: Line 114, the meaning of the sentence is not clear

Among ROS H2O2 is most stable and therefore is the most important component in signaling [19].

Response 2: Reframed and highlighted in green

Point 3:

Line 137, the acronym ER was not written in full

Line 212, there is a repetition of the word "function"

Line 137, the acronym PUFA was not written in full

Line 251, “GSSO” should be modified in “GSSG”

Line 256, “non-photosynthetic and non-photosynthetic” should be modified in “photosynthetic and non-photosynthetic”

Line 261, “moment” should be modified in “property”

Response 3: All the suggestions are incorporated and highlighted in green in the revised manuscript for your kind look

Point 4: Line 269-273, there are no spaces between words

Response 4: May be software problem as it is clear  in my computer and not able to understand what should I do. Pls guide

Point 5 : Line 280, the localization of Fe-Sod should be specified

Response 5: I think it is more detailing about the Class I, II and III (Monofunctional heme containing catalases, heme containing catalase-peroxidases and manganese containing catalases), should we write these details in text.

Point 6: Line 298, unify the wording: class 2 or class II

Response 6: Corrected and highlighted in green

Point 7: Line 311-330, there are no spaces

Response 7: Now the revised manuscript is in pdf file so that MS Word problem may be eliminated.

Point 8:

Line 332, the acronym AA was not written in full

Line 339-349, there are no spaces between words

Line 428, “GSSG/GSH” should be modified in “GSH/GSSG”

Line 485, “Innterestingly” should be modified in “Interestingly”

Response 8: All the suggestions are incorporated and highlighted in green in the revised manuscript for your kind look.

Point 9: Line 486, what is the meaning of “cassava”?

Response 9: Incorporated as suggested, (Manihot esculenta commonly called cassava) highlighted in green

MAJOR REVISIONS

Point 10: Regarded as the exceptional ability of flavonoids to counteract the damage related to the increase of ROS, I suggest to improve paragraph 3.1.5.

Response 10: Incorporated as suggested and highlighted in blue in main manuscript

Flavonoids are the only antioxidant biomolecules which possess the capacity to absorb UV radiation. Absorbed energy quanta will result a generation of ROS. The ROS generation from certain flavonoids was studied using fluorescence probes. Flavonoids generate three ROS types: the superoxide anion radical (O2 •−), the hydroxyl radical (• OH), and singlet oxygen (1O2). This is based on the presence of the 2,3 double bond found in all flavonoids. In this context the finding of Jiang et al (2019) is noteworthy to mention: These authors have published that natural flavonoids are not able to show the ROS production activities. This activity rather is exerted by derivaties of quercetin like 3’,4’,5,7-tetra-O-methylquercetin. The authors have shown that this derivative is able to generate under exposure to UV and X ray irradiation (• OH) and (1O2) due to the 2,3 double bond and the 3-OH group [56].

Point 11: I find very interesting the observation regarding the similarity of the stress perception and its consequent signaling between plant and animal cells. Could the authors provide more information about the pathways involved and their consequences? I might suggest adding the following reference: (doi: 10.1007/s12079-012-0173-3), in order to discuss also these aspects.

Response 11: Incorporated the references as suggested by reviewer. Thanks a lot for kind suggestions.

It was an important and quite surprising finding that there are similarities in stress responses of plant and animal cells. Heavy metals are well known for their adverse effects on cell homeostasis in cells of plants and animals. This allows discussing general concepts underlying stress tolerance in cells differing in physiology and structure as much as plant and animal cells. We will mention here signaling events that have been observed subsequent to exposure to cadmium, for example. Most obviously we can find common signaling components having similar functions in both cell types. The most important ones are MAP kinases and ERK kinases (extracellular signal regulated kinases). These kinases are involved in regulation of important functions such as the control of  cell division. Disturbance of such pathways may lead to cell death or uncontrolled cell division as observed in tumors, for instance. Signal transduction can be initiated by Cd sensing and subsequent release of signaling molecules such as Ca2+, ROS and NO. On the other hand, significant differences have been found with respect to the compounds produced subsequent to signaling activated gene expression, such as specific animal and plant hormones. Thus, the general concept of Cd sensing and initial response may be evolutionary conserved [112]. This assumption is supported by more detailed observations. Both, in animal and plant cells, ROS production is catalyzed by a Cd activated NADPH oxidase [113].

Point 12: Moreover, it might be interesting to point out similarities and differences in the action of ROS on ion channels of plants and animals. I might suggest adding the following reference: (doi: 10.1242/jcs.00201; doi: 10.1111/j.1365-313X.2006.02971.x; doi: 10.1111/apha.13796), in order to discuss also these aspects.

Response 12: incorporated as suggested and highlighted in text.

Further, it became evident from the experiments of Demidchik et al., 2003, 2007[114, 115] that cell signaling is mediated by free radical and redox mechanisms. This signaling is targeting ion channels and receptor complex systems that will start a series of responses regulating cellular metabolism. During the last decades, it became evident that in plants as well as in animals the production and an increased content of ROS is triggering signaling pathways. These signaling events initiate responses as different as hormone synthesis geotropism and stress responses. In a similar way Ca2+ signaling is involved in several signaling sequences. ROS stimulated opening of Ca2+ channels can result an amplification of stress signals. Moreover, involvement of both signaling molecules in several signaling pathways is understood to be the reason for observed cross reactions and reciprocal modulation of responses when several stresses are occurring simultaneously.

Reviewer 2 Report

The discussed manuscript "Metabolic Pathway of Natural Antioxidants, Antioxidant Enzymes and ROS providence" is interesting and summarizes much publicly available and widely known information on oxidative stress. The topic chosen by the authors is very extensive, and could even be a book, so I understand that the authors had to make decisions about which information to include in the manuscript and which to omit. However, in this way some chapters on particular issues are described very briefly, e.g. very poorly described ROS production caused by abiotic stress. I suggest you think again about the individual chapters and supplement them, for example, with the most important issues related to them.

Also:

  1. In chapters 3.1.5, 3.2.3 and 3.2.5 the text fused together (no spaces), which is a big problem to read it. This needs to be corrected.
  2. I suggest you re-read the paper in order to eliminate minor mistakes, eg punctuation.
  3. It is necessary to standardize the way of presenting individual references in the section references.
  4. The manuscript lists drawings that are supplementary materials, but I do not have access to this document so I cannot refer to it. Besides, instead of describing these pictures exactly in the legend, it could be woven into the text of the manuscript, and only the pictures in supplementary materials for a continuity (unity) of what in supplementary and the text in the manuscript.
  5. There is only one table at manuscript and the drawings are placed in suppl. Why? The manuscript diversified with drawings could be even more interesting and eye-catching.

Author Response

Dear Sir,

Kindly find enclosed herewith the reviewers suggestion and incorporation as per our best efforts. The authors are grateful to the reviewer for the following important suggestions.

Point 1:

In chapters 3.1.5, 3.2.3 and 3.2.5 the text fused together (no spaces), which is a big problem to read it. This needs to be corrected.

Response 1: 

I am sorry to state that in our text it is showing clear may be the MS office software of ours is causing the problem so I am submitting the revised MS in pdf format for your kind look.

Point 2: I suggest you re-read the paper in order to eliminate minor mistakes, eg punctuation

Response 2: Corrected as suggested and highlighted in green 

Point 3:

It is necessary to standardize the way of presenting individual references in the section references.

Response 3:Corrected as suggested

Point 4: 

The manuscript lists drawings that are supplementary materials, but I do not have access to this document so I cannot refer to it. Besides, instead of describing these pictures exactly in the legend, it could be woven into the text of the manuscript, and only the pictures in supplementary materials for a continuity (unity) of what in supplementary and the text in the manuscript.

Response 4: Drawings are incorporated in the text and the pdf of updated Manuscript is submitted for your kind look.

Point 5:

There is only one table at manuscript and the drawings are placed in suppl. Why? The manuscript diversified with drawings could be even more interesting and eye-catching.

Response 5: All the Figures 1 – 7 are incorporated in the text for your kind reference.

Round 2

Reviewer 1 Report

The authors have satisfyingly addressed all concerns and suggestions. However, I still suggest to review the Figure 6 B. After this modification, the following manuscript could be accepted for the publication.

Reviewer 2 Report

After the corrections are made, the manuscript can be published in its current form